# A Prospective Observational Study on Analyzing Lung Cancer Gene Mutation Variant Allele Frequency (VAF) and Its Correlation with Treatment Efficacy

**DOI:** 10.3390/ijms252111694

**Published:** 2024-10-30

**Authors:** Yusuke Shinozaki, Kei Morikawa, Hirotaka Kida, Hiroshi Handa, Hisashi Saji, Seiji Nakamura, Yoshiharu Sato, Yumi Ueda, Fumihiko Suzuki, Ryo Matoba, Masamichi Mineshita

**Affiliations:** 1Division of Respiratory Diseases, Department of Internal Medicine, St. Marianna University School of Medicine, Kawasaki 216-8511, Japan; 2Department of Chest Surgery, St. Marianna University School of Medicine, Kawasaki 216-8511, Japan; 3DNA Chip Research Inc., Kawasaki 211-0004, Japan

**Keywords:** cytology specimen, gene panel analysis, next-generation sequencing, non-small cell carcinoma (NSCLC), variant allele frequency (VAF)

## Abstract

A few studies have reported on the variability of the variant allele frequency (VAF) of gene mutations and the relationship between VAF and the therapeutic effect of molecular-targeted drugs. This joint study was conducted from May 2020 to January 2022 between St. Marianna University School of Medicine and the DNA Chip Research Institute. Cytology samples were used to verify the usefulness of a lung cancer compact panel test, which is a high-sensitivity next-generation sequencing (NGS) gene panel test. We analyzed the distribution of VAF and the duration of the initial molecular-targeted drug administration in patients with advanced-stage epidermal growth factor receptor (*EGFR*) mutations. Of the 196 patients diagnosed with non-small cell carcinoma (NSCLC), gene mutations were detected in 114 (58.2%) cases and in 68.7% of patients with adenocarcinomas (an increased detection rate). A VAF of 30% or more for DNA mutations was confirmed in 35 patients (33.7%), 10% to <30% in 38 (36.5%), and <10% in 31 (29.8%), including 18 patients (17.3%) with <6%, which is considered the lower limit of the LOD for other companion diagnostic kits. *EGFR* mutations were detected in 59 patients (30%), of whom 35 were in an advanced stage and received molecular-targeted drugs as their initial treatment. Groups A and B comprised patients with a VAF of ≥10% and <10%, respectively, with 27 patients (median VAF 30.6%, range 11.4–77%) in group A and 8 (median VAF 5.4%, range 0.1–8.5%) in group B. The duration of the administration of molecular-targeted drugs was 17 months (range 3–42 months) in group A and 14 months (range 1–45 months) in group B, with no statistically significant difference between the two groups (*p*-value = 0.7). Twenty-seven percent of patients had a VAF < 10%. Even in patients with a low VAF for gene mutations, sufficient therapeutic effects were observed with molecular-targeted drugs, suggesting the importance of detecting mutations using a highly sensitive method at initial diagnosis. Further prospective validation studies of a larger number of cases are warranted.

## 1. Introduction

Since personalized medicine for lung cancer based on genetic mutation information has become more widespread, accurate gene panel testing at the initial diagnosis stage has become crucial [1,2,3]. The companion diagnostics (CDx) currently available for detecting gene mutations are diverse, including multi-plex and single-plex tests [4]. However, large differences in their performance in terms of detection, sensitivity, and mutation variants have been encountered in clinical settings [5,6]. In particular, cases of false negatives due to differences in the detection sensitivity (limit of detection [LOD]) of gene mutations in each test method have been reported [7]. These false negatives have resulted in missed treatment opportunities, which pose a major problem for patients. Although few studies suggest that treatment with molecular-targeted drugs is effective, even when variant allele frequency (VAF) is low, evidence supporting the significance of highly sensitive mutation detection is lacking [8,9,10]. The accurate detection of variant allele frequency (VAF) plays a critical role in the effective administration of targeted therapies for lung cancer. High-sensitivity gene panel testing ensures that even low-frequency mutations are identified, potentially expanding treatment options for patients who might otherwise be overlooked by less sensitive methods. Therefore, in this study, we used a lung cancer compact panel (LCCP) [11,12], a highly sensitive next-generation sequencing (NGS) method that uses cytology samples, to analyze the degree of variation in the VAF of mutations between patients, in order to estimate the degree of false negatives occurring in real-world clinical practice and analyze whether VAF and treatment efficacy are correlated.

## 2. Results

Of the 196 patients diagnosed with non-small cell carcinoma (NSCLC), 71 were male (36.2%) and 125 were female (63.7%). The median patient age was 72 years (range 26–90 years). Bronchoscopy was the most common examination method (80%), followed by percutaneous biopsy. The most common histological type of NSCLC was adenocarcinoma (78.6%), followed by squamous cell carcinoma (16.8%) (Table 1). Gene mutations were detected in 114 cases (58.2%) of the patients with NSCLC (Figure 1A), and the detection rate increased to 68.7% when adenocarcinoma was considered in the overall population.

The VAF of DNA mutations was ≥30% in 35 patients (33.7%), 10% to <30% in 38 (36.5%), and <10% in 31 (29.8%), including 18 patients (17.3%) with <6%, which is considered the lower limit of the LOD for other companion diagnostic kits (Figure 1B). No particular tendency in the VAF values was observed depending on the type of gene mutation or variant. In the case of *EGFR* mutation variants, all four cases of mutations were not covered as detectable variants according to the product description of any of the other genetic tests: AmoyDx did not detect mutations in any of the four cases, ODxTT did not in three, and cobasDx did not in one (Table 2).

Of the patients in whom *EGFR* mutations were detected, 35 were in the advanced stage and were administered molecular-targeted drugs as their initial treatment. Group A comprised patients with a VAF of ≥%10 and group B comprised patients with a VAF of <10%, with the following distributions: 27 (median VAF 30.6%, range 11.4–77.8%) in group A and 8 (median VAF 5.4%, range 0.1–8.5%) in group B. The treatment regimen administered to group A was osimertinib/afatinib/erlotinib/gefitinib/erlotinib + ramucirumab (15/7/2/2/1) and that administered to group B was osimertinib/afatinib/erlotinib + ramcirumab (4/3/1). The medium administration period of the molecular-targeted drugs was 17 months (range 3–42 months) and 14 months (range 1–45 months) in groups A and B, respectively, with no statistically significant difference between the two groups with regard to the data cutoff in May 2024 (*p*-value = 0.7) (Figure 2).

## 3. Discussion

In this study, we used LCCP, a highly sensitive NGS, to demonstrate the variation in the VAF of gene mutations among patients. This prospective study demonstrated that a certain number of patients have a VAF of <10%. Furthermore, we demonstrated that a certain degree of therapeutic effect could be guaranteed with molecular-targeted drugs, even in patients with a VAF of <10%. Therefore, the detection of gene mutations with high sensitivity is important to expand the scope of personalized treatment.

Currently, lung cancer gene panel testing, using ODxTT and Amoy Dx as PCR methods, is widely used ^4.13^. Recently, there have been concerns about the declining detection rate of *EGFR* gene mutations, with a large disparity observed between facilities. The Japan Lung Cancer Society conducted a nationwide survey of cancer center hospitals (345 facilities) with a focus on the detection rate of *EGFR* gene mutations [13]. In NSCLC, the detection rate of *EGFR* mutations is approximately 35–40% [12,14]; notably, the detection rate differs depending on the type of gene panel test used, the sample type, and facility. Although no further scientific pursuit has been reported, a large difference in the detection sensitivity (LOD) of panel tests and mutation variants has been observed [15]. In particular, it has been suggested that ODxTT (LOD range: 4.4–5.3%), single-plex PCR testing, and cobasDx (LOD range: 1.26–6.81%) may present false negatives in cases of a VAF of <6% [4,16]. It is noteworthy that in our study, 17% of the patients had a VAF of <6% for DNA mutations. In addition, gene mutations that could only be detected by LCCP were confirmed as *EGFR* mutation variants in four patients (6.8%). From the above perspective, it was suggested that in clinical practice, when panel tests such as ODxTT and AmoyDx, or single-plex tests such as cobasDx, are used, false negatives are obtained in at least 10–15% of cases.

The significance of this study is that, although limited to DNA mutations, the VAF of gene mutations varies greatly by case. Although the number of cases is small, especially for rare mutations such as *MET* exon 14 skipping, *BRAF*, and *ERBB2* mutations, no consistent trend in VAF values was observed for any gene mutation. The number of tumor cells and the VAF vary depending on the case, which might be due to intratumor heterogeneity and the microenvironment surrounding the tumor cells, especially when pleural effusions contain large amounts of inflammatory cells such as lymphocytes. Because the number of malignant cells that can be obtained varies, depending on the examination and the tumor itself, our results suggest that it is important to use a gene testing method that is as sensitive as possible so that the results are reliable even when the number of tumors is small. From this perspective, LCCP can be said to be a very promising gene panel test.

Reports on VAF and treatment efficacy are scarce. Few reports showed a difference in the treatment efficacy of osimertinib depending on the VAF of the *T790M* mutation in a re-biopsy after treatment for *EGFR*-mutant lung cancer [8,9,10]. In our study, although the number of advanced-stage cases was limited, the results showed that even in the group of patients with a low VAF of <10%, there was no significant difference in the treatment period compared to the group of patients with a VAF of ≥10%. Clinical trials have reported that the duration of the response to *EGFR*-tyrosine kinase inhibitors (TKIs) in patients with *EGFR* mutations exceeded 12 months. Similarly, the median TTF of 14 months observed in this study was comparable to these results [17,18,19]. It is also important to note that there is a possibility that the median TTF may be extended if the observation period is extended in the future. These findings underscore the clinical importance of detecting low-frequency mutations using highly sensitive methods. In cases where the VAF is below the detection threshold of standard companion diagnostics, patients may miss the opportunity for effective targeted therapies. By utilizing LCCP, which demonstrates enhanced sensitivity, our study suggests that patients with VAFs below 10% can still achieve substantial therapeutic benefit. This highlights the need for integrating high-sensitivity NGS panels in routine clinical practice to ensure that all patients receive optimal therapeutic strategies based on their specific genetic profiles. Comprehensive genomic profiling (CGP) panels can identify potential driver mutations even after the onset of acquired resistance, offering opportunities for targeted treatment adjustments [1,3]. However, studies have shown that if the optimal treatment is not administered during first-line therapy, the overall effectiveness of subsequent treatments can be significantly reduced [3]. Administering the most effective therapy early on maximizes tumor response and may delay the emergence of resistance mutations, thus leading to better patient outcomes. This underscores the need for high-sensitivity testing at the initial diagnosis to ensure that patients receive the best possible therapy from the outset, minimizing the chance of resistance-driven progression and maintaining longer-term disease control. Furthermore, there is no knowledge as to which drug is the most effective when VAF is low, and prospective studies are needed in the future.

The first limitation of this study is that the NGS analysis and VAF measurements were performed using cytology samples. However, it has been reported that there is sufficient correlation between tissue and cell samples [12]. LCCP allows for the detection and analysis of cytological samples with a low tumor content or a small number of absolute tumor cells. Therefore, detection opportunities are expected to increase, especially when there is insufficient biopsy tissue or when only a liquid sample can be obtained. Second, this study had a small cohort. Therefore, further prospective studies with a multicenter design are required for further verification of the correlation between VAF and treatment efficacy. In addition, progression-free survival is a more accurate measure of treatment efficacy, but the imaging evaluation method was not defined in this study; therefore, TTF was used for evaluation.

## 4. Materials and Methods

### 4.1. Aim and Study Design

The purpose of this study was to clarify the distribution of VAF for each gene mutation and analyze the correlation between VAF and treatment efficacy.

In this single-center prospective study, we evaluated the mutation cell profile constructed using LCCP results and VAF distributions. In addition, the duration of administration was analyzed for patients with advanced-stage *EGFR* mutations who were administered molecular-targeted drugs. This study was conducted in accordance with the Declaration of Helsinki and approved by the Institutional Review Board of St. Marianna University School of Medicine (approval number 4814). Written informed consent was obtained from all patients.

### 4.2. Patient Selection

Consecutive patients who underwent diagnostic procedures between May 2020 and January 2022 were enrolled. Patients with suspected lung malignancies who required a pathological diagnosis, were 20 years of age or older, and provided written consent to participate in the study were eligible. The included patients were suspected of having lung cell carcinoma based on computed tomography (CT) or positron emission tomography-CT imaging.

### 4.3. Diagnostic Procedures

Endobronchial ultrasonography (EU-ME2; Olympus, Tokyo, Japan) was used for bronchoscopic examinations with or without a guide sheath kit (Olympus). Endobronchial ultrasonography-guided transbronchial needle aspiration (EBUS-TBNA) or endoscopic ultrasound-guided fine-needle aspiration was performed using a flexible fiberscope, with two to three passes performed using a 22-gauge needle. A CT/ultrasound (US)-guided core needle biopsy was performed thrice using a semiautomatic aspiration device (Temno Evolution, Care Fusion Japan, Tokyo, Japan) with a 20-gauge 11 or 15 cm needle. Additionally, a sufficient number of tissue specimens were collected, if available.

### 4.4. Cytology Specimen Collection

Transbronchial biopsy samples were prepared by adhering the cells, collected by scraping the lesions with a brush, to a glass slide, and stirring in 4 mL of normal saline at least 2–3 times. Needle aspirations and biopsies were performed to remove the core tissue for pathological evaluation. The needle was then washed with approximately 1 mL of normal saline and flushed with air at least 2–3 times before sample collection. The washing solution was evenly divided into two containers, one for the sample container and the other for cytological evaluation (paired cytology samples). If no malignant cells were found in the paired cytological samples, the patient was excluded from the second registration. Liquid pleural effusion specimens (at least 20 mL) were collected, divided into two containers, and centrifuged. The pellets were then stored in a sample container for pathological evaluation.

### 4.5. Mutation VAF Analysis Using Lung Cancer Compact Panel^TM^ (LCCP)

Cytological specimens were processed with the Maxwell^®^ RSC Blood DNA Kit and Maxwell^®^ RSC simply RNA Cells Kit (Promega, WI, USA). For the FFPE specimens, DNA and RNA were purified using the Maxwell^®^ RSC DNA FFPE Kit and Maxwell^®^ RSC RNA FFPE Kit (Promega, WI, USA) according to the manufacturer’s protocol. Using purified nucleic acid, a lung cancer compact panel (LCCP: DNA Chip Research Inc., Tokyo, Japan) NGS assay was performed as described previously [11]. The LCCP is an amplicon-based high-sensitivity NGS panel capable of measuring eight druggable genes (EGFR, BRAF, KRAS, ERBB2, ALK, ROS1, MET, RET) for lung cancer. The experimental process is briefly described below. As an initial analysis, 5 ng of DNA (double-stranded) and 5 ng of RNA were used for the assay of each module. Analytical validation was conducted for assays using the same input amount, and proficiency was confirmed [11]. Therefore, 10 ng of dsDNA and 10 ng of RNA were set as the minimum requirement for the yield of purified nucleotides. The QubitTM fluorometer (Thermo fisher, Waltham, MA, USA) with dsDNA HS (High Sensitivity) Assay Kits and NanoDrop^®^ UV-spectrophotometry (Thermo fisher, Waltham, MA, USA) were used for the quantification of genomic DNA and total RNA, respectively. The TapeStation (Agilent, Santa Clara, CA, USA) Genomic DNA assay was used for the assessment of the quality of purified DNA (DIN: DNA Integrity Number). The TapeStation (Agilent) RNA HS assay or Bioanalyzer (Agilent) was used for the assessment of the quality of purified RNA (RIN/eRIN: RNA Integrity Number and DV200%). For the DNA assay, multiplex PCR using KOD-Plus-Neo (Toyobo, Osaka, Japan) was performed to amplify EGFR (exon 18–21), BRAF (exon 15), KRAS (exon 2), ERBB2 (exon 8, 17, 20), and MET (near exon 14). Two DNA panels (DNA module1 and DNA module2) were designed and optimized to detect somatic mutation sensitively and quantitatively, with the unbiased amplification of these hotspot regions. Amplicon-based library construction was performed as described previously [11]. Sequence data were acquired using MiSeq (Illumina, CA, USA) for the constructed sequence library (2 *×* 150 bp). During data analysis, the Illumina adapter sequences were trimmed using Trimmomatic v0.33 and paired-end sequences were joined using the FLASH v1.2.11 fastq joining tool. The joined reads were then mapped on the target regions of the human genome using BWA aligner v0.7.17 and the mutation variant was identified by analyzing bam format alignment output using custom programming scripts. For the assessment of assay quality, 5000-read depths for DNA module1 and 2000 read depths for DNA module2 were set as the minimum threshold of the sequencing read pairs of each amplicon region. For the RNA module, 300 sequencing reads of the internal Hprt1 amplification were set as a minimum threshold. Insufficient amplification below the coverage threshold were judged as assay failure. The analytical performance of LCCP was thoroughly validated according to the ICH guidelines (https://www.pmda.go.jp/files/000156867.pdf accessed on 1 August 2024). 

### 4.6. Outcome Assessments: Statistical Analysis

A log-rank test was performed to assess whether there was a significant difference in the treatment effect (TTF) of EGFR-TKI between allele frequencies below 10% and those above or equal to 10%. A significance level of 5% was set for this analysis. The survival analysis was conducted by R version 4.4.1.

## 5. Conclusions

In conclusion, this study demonstrated the clinical significance of detecting gene mutations with a high-sensitivity NGS panel and administering corresponding molecular-targeted drugs. In the future, it will be necessary to verify this finding prospectively in a larger population.

## Figures and Tables

**Figure 1 ijms-25-11694-f001:**
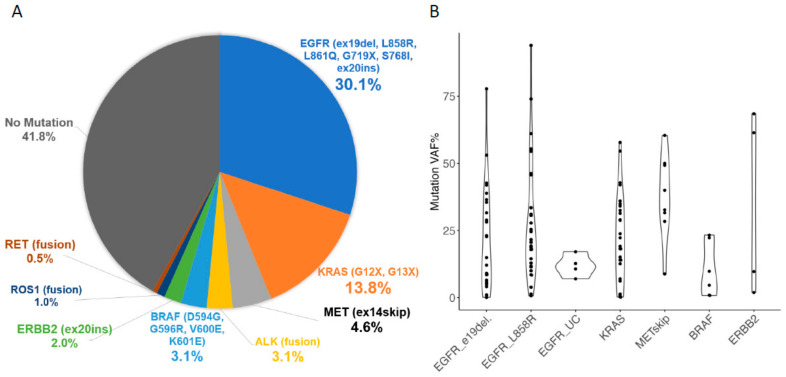
Pie chart of mutation cells detected by LCCP assay for non-small cell lung carcinoma (**A**) and each mutation VAF% (**B**). *EGFR*, epidermal growth factor receptor; e19del, exon 19 deletion; L858R, exon21 L858R point mutation; LCCP, lung cancer compact panel; UC, uncommon mutation; VAF, variant allele frequency.

**Figure 2 ijms-25-11694-f002:**
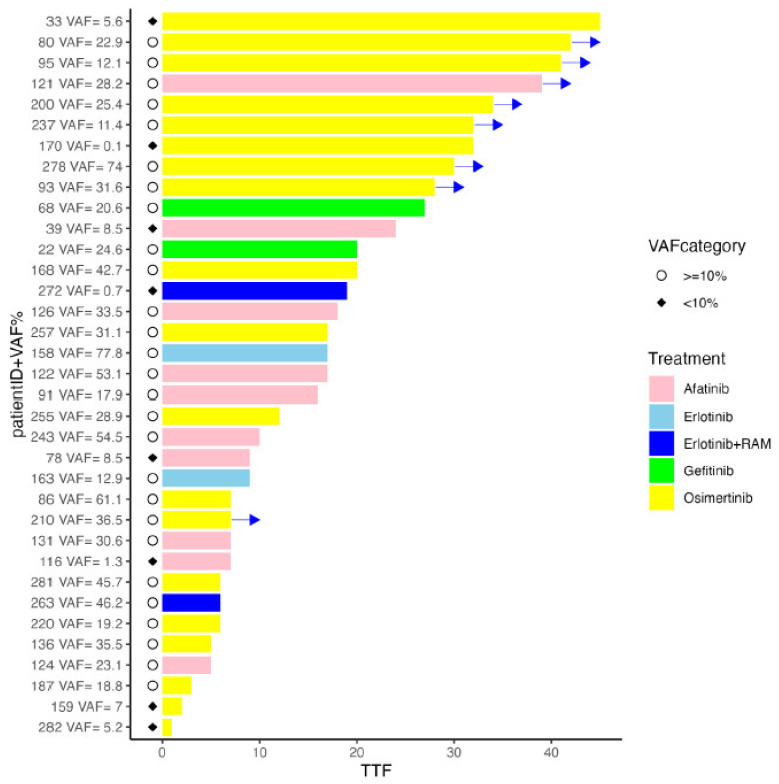
Swimmer’s plot of time to treatment failure according to VAF% for *EGFR*-mutant cases. *EGFR*, epidermal growth factor receptor; RAM, ramucirumab; TTF, time to treatment failure; VAF, variant allele frequencies; →, Treatment is ongoing.

**Table 1 ijms-25-11694-t001:** Patient characteristics and LCCP analysis.

Patient Charasteristics	*n* = 196	(%)	Pathological Diagnosis	Cases	(%)
Sex			Adenocarcinoma	154	(78.6)
Male	71	(36.2)	Squamous cell carcinoma	33	(16.8)
Female	125	(63.7)	Not otherwise specified, other	9	(4.6)
Median age (range)	72	(26–90)	**LCCP Mutation Detection**	**Cases**	
			*EGFR*	59	
			*KRAS G12X G13X*	27	
**Diagnostic Procedure**			*ALK*	6	
EBUS-TBB	135	(68.9)	*BRAF*	6	
EBUS-TBNA, EUS-FNA	25	(12.8)	*MET exon14 skipping*	9	
CT/US-guided puncture	15	(7.7)	*RET*	1	
Pleural effusion	5	(2.6)	*ERBB2*	4	
Other	16	(8.2)	*ROS1*	2	

CT, computed tomography; EBUS-TBB, endobronchial ultrasonography-guided transbronchial brushing; EBUS-TBNA, endobronchial ultrasonography-guided transbronchial needle aspiration; EUS-FNA, endoscopic ultrasound-guided fine-needle aspiration; US, ultrasound; LCCP, lung cancer compact panel; US, ultrasound.

**Table 2 ijms-25-11694-t002:** Four cases of false negatives using other testing methods.

Sample ID	Histological Type	*EGFR* Variant (ODxTT, AmoyMulti, Cobas Not Supported)	Mutation Call	VAF Percentage
60	Ad	ODxTT-, Amoy-, Cobas+	*EGFR*_p.E746_T751delinsIP	8.6
67	Ad	ODxTT-, Amoy-, Cobas+	*EGFR*_p.E746_T751delinsIP	1
110	Ad	ODxTT+, Amoy-, Cobas+	*EGFR*_p.A767_V769dup	12.7
134	Ad	ODxTT-, Amoy-, Cobas-	*EGFR*_p.E746_A750delinsQP	6.2

Ad, adenocarcinoma; Amoy, Amoy Dx lung cancer multi-PCR gene panel; Cobas, cobas EGFR Mutation Test v2; EGFR, epidermal growth factor receptor; ODxTT, Oncomine Dx Target Test Multi-CDx system; VAF, variant allele frequency.

## Data Availability

Detailed genetic analysis data will be disclosed upon request.

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
