# Peer review of "A Prospective Observational Study on Analyzing Lung Cancer Gene Mutation Variant Allele Frequency (VAF) and Its Correlation with Treatment Efficacy"

_ijms, 2024, doi:10.3390/ijms252111694_

Round 1
Reviewer 1 Report
Comments and Suggestions for Authors
The aims of this study were to examine the distribution of variant allele frequency (VAF) for individual gene mutations detected in Transbronchial biopsy samples collected from patients with EGFR mutant NSCLC based on the high-sensitivity next generation sequencing (NGS) results and to analyze the correlation between VAF and treatment efficacy. A total number of 196 patients with NSCLC were evaluated using NGS. Percent of patients with VAF > 10% and < 10% was 70.2% and 29.8%, respectively. Of the 35 patients who had advanced NSCLC and treated with protein kinase inhibitors, 27 patients had VAF > 10% (with the median VAF of 30.6%) and 8 patients had VAF < 10 % (with the median VAF of 5.4%). The results showed that there was no statistically significant difference in time to treatment failure between the VAF-no-less-than-10% group and VAF-less-than-10% group (17 months versus 14 months).
The study did not produce any insightful information. The results did not support the conclusion that there is a "clinical significance of detecting gene mutations with high sensitivity NGS panel", which results may be used to guide the use of molecular-targeted drugs". The conclusion was not convincing when the sample size was small.
Comments on the Quality of English Language
Moderate editing of English language required.
Author Response
Dear reviewer
Thank you for your kind comment. I have revised the conclusion that can be drawn from this very small number of cases. Although it is true that this is a study of a small number of cases from a single institution, I am convinced that there are almost no reports of studies that prospectively examine the relationship between gene mutation allele ratio of cytology samples and the treatment effect of molecular-targeted drug. In addition, I have asked editage to proofread the English text and have them correct it in a native language.
Reviewer 2 Report
Comments and Suggestions for Authors
The manuscript entitled A Prospective Observational Study on Analyzing Lung Cancer Gene Mutation Variant Allele Frequency (VAF) and Its Correlation with Treatment Efficacy represents a timely relevant manuscript requiring moderate comments to be accepted for the publication on this journal
- In the introduction section, please concisely overview the opening challenges in molecular testing of predictive biomakrers in lung cancer patients?
- In the methodological section, please, could the authors define criteria for establishing mentioned MAF cut of?
- In the methodological section, please, could the authors add a brief description of technical parameters of NGS panel?
- In the study design section, please, could the authors consider the interaction between TKIs and molecular alteration?
- In the study design section, please, could the authors discuss about p.G12X? How this result may impact on the molecular result?
- In the results section, please, could the authors distinguish among different TKIs?
- Please, could the authors perform moderate native english revision?
Comments on the Quality of English LanguageModerate native english revision
Author Response
Dear reviewer
Thank you for your kind comment. We have looked into your question in detail and are reviewing our paper.
- In the introduction section, please concisely overview the opening challenges in molecular testing of predictive biomakrers in lung cancer patients?
Answer: As potential problems with gene mutation searches, we have already mentioned the difference in sensitivity and target variants. As you pointed out, there are various other problems, but we believe that this study is sufficient as an introduction to research focusing on sensitivity and variants.
- In the methodological section, please, could the authors define criteria for establishing mentioned MAF cut of?
Answer: The VAF LOD definition has already been described in the method as follows:
with LODs of 0.14%, 0.20%, 0.48%, 0.24%, and 0.20% for driver mutations such as EGFR exon-19 deletion, L858R, T790M, BRAF V600E, and KRAS G12C, respectively. The basic technology for achieving high sensitivity and analytical validation of LCCP has been reported previously. 11
- In the methodological section, please, could the authors add a brief description of technical parameters of NGS panel?
Answer: The detailed methodology is presented in Reference 11 and has already been published in Cancers, a sister journal of MDPI, and is available in Reference 12.
- In the study design section, please, could the authors consider the interaction between TKIs and molecular alteration?
Answer: I'm very sorry, but I don't fully understand the intent of your question. About T790M resistant mechanism or MET amplification during the exposure of EGFR-TKI? This study only observed the effects of first-line treatment and did not conduct any further molecular pathological examinations.
- In the study design section, please, could the authors discuss about p.G12X? How this result may impact on the molecular result?
Answer: The number of KRAS cases in Japan is low compared to Europe and the US, and KRAS G12C, which is the target of molecular targeted drugs, is even rarer. Therefore, detailed studies are difficult because it is almost impossible to evaluate the effectiveness of treatment.
- In the results section, please, could the authors distinguish among different TKIs?
Answer: This is an important point, and specific treatments are listed in the results figure. However, because the number of cases is small, we are unable to determine which treatment is appropriate. However, this is an important point, and we have added it to the discussion.
- Please, could the authors perform moderate native english revision?
Answer: The English has already been proofread by a native English speaker at editage.
Round 2
Reviewer 1 Report
Comments and Suggestions for Authors
Line 134 and Line 135. Since there was only one subsection under Section 2.6, the authors should consider remove Subsection 2.6.1.
Line 178-181. What statistical analysis was performed to compare between Group A and B? The statistical analysis should be described in Section 2.6.
Comments on the Quality of English Language
Minor editing is needed.
Author Response
Line 134 and Line 135. Since there was only one subsection under Section 2.6, the authors should consider remove Subsection 2.6.1.
Line 178-181. What statistical analysis was performed to compare between Group A and B? The statistical analysis should be described in Section 2.6.
Answer: We revised above point in the submitted manuscript.
Reviewer 2 Report
Comments and Suggestions for Authors
No other comments
Author Response
No comments and no need to answer.